# Transient Magnetic Properties of Non-Grain Oriented Silicon Steel under Multi-Physics Field

**DOI:** 10.3390/ma15238305

**Published:** 2022-11-23

**Authors:** Anqi Wang, Baozhi Tian, Yulin Li, Shibo Xu, Lubin Zeng, Ruilin Pei

**Affiliations:** 1School of Electrical Engineering, Shenyang University of Technology, Shenyang 110870, China; 2Suzhou Inn-Mag New Energy Ltd., Suzhou 215000, China

**Keywords:** non-grain oriented silicon steel, multi-physics field, high-performance motors, efficiency

## Abstract

High-speed, high-efficiency and high-power density are the main development trends of high-performance motors in the future. At present, the design accuracy of traditional electric machines is already high enough; however, for the future demand of high performance and utilization in special environments (such as aviation and aerospace fields), more thorough research of materials’ performance under multi-physics field (MPF) conditions is still needed. In this paper, a test system that combined temperature, stress and electromagnetic fields along with other fields, at the same time, is built. It can accurately simulate the actual complex working conditions of the motor and explore the dynamic characteristics of non-grain oriented (NGO) silicon steel. The rationality of this method is verified by checking the test result of the prototype, and the calculation accuracy of the motor model is improved.

## 1. Introduction

Silicon steel is one of the most important materials in motor. Under the actual working state of the motor, the core material is in a physical field where temperature, stress, frequency conversion and other external fields are coupled at the same time. For example, when the built-in permanent magnet synchronous motor is running at high speed, the alternating frequency of the magnetic field in iron core is high [1], and the iron loss increases exponentially as the frequency increases [2]. In addition, high frequency will also cause skin effects on the surface of silicon steel sheet, which will further increase the iron loss [3]. The loss of certain grades of silicon steel will increase several times when giving them compressive stress from 0 MPa to 100 MPa [4,5]. Silicon steel material will appear as a dynamic strain aging phenomenon under the action of temperature [6], and its tensile strength will change at the same time. Similarly, the heat dissipation problem and demagnetization problem caused by temperature rise during the operation of permanent magnet synchronous motor indicate that it is necessary to consider the influence of frequency, stress and temperature on the core material.

Traditional motor design and finite element analysis are calculated iteratively through the basic data of materials under power frequency, no external force and constant temperature conditions. However, with the improvement of the requirements for high-performance motors in the future, how to accurately calculate and predict the performance of motors under extreme conditions is particularly important. The magnetic properties and mechanical properties of silicon steel under MPF have gradually become a research hotspot. In Reference [7], the author explored the influence of stress and temperature on the iron loss of silicon steel, and completed the modification of the calculation coefficient of iron loss according to the influence.

Many scholars have studied the effect of a single physical field (SPF) change on silicon steel sheets’ magnetic properties. In addition to the excitations, the microstructure of silicon steel will change with the change in stress [8,9], thus causing effects to its magnetic properties. When it is applied with external force, silicon steel will show magneto elastic properties [10] and stress demagnetization [7], which will also promote or deteriorate the magnetization of silicon steel. With the increase in tensile stress, most of the silicon steel will show a trend that its magnetic density firstly increases and then decreases. However, the effect of stress on silicon steel sheet loss is not infinite. When stress reaches a certain value, the loss will remain in a relatively stable numerical range [11].

By applying the temperature field to Epstein coils, Chen’s team explored the influence of temperature rise on the magnetic properties of soft magnetic materials, and found that the magnetic density of materials decreased with the increase in temperature [12]. The influence of temperature on the electromagnetic properties of 0.5 mm medium grade NGO silicon steel was studied, and the action mechanism model was proposed [13]. Temperature coefficient K and effective resistivity of silicon steel lamination was subsequently introduced to explore the change trend of iron loss with temperature, and came to a conclusion that the iron loss decreases with temperature rise [14].

In 2016, American scholar G.M.R. Negri [15] conducted experiments involving thermal, metallurgical and electromagnetic measurements on silicon steel, simulated the evolution process of loss using the scalar hysteresis model and predicted the aging of silicon steel based on the Arrhenius formula, providing important significance for motor design. Similarly, Andreas Krings and Oskar Wallmark studied the influence of motor operating temperature on its magnetic performance and proposed a core loss model including temperature influence coefficient [16]. After them, different researchers continuously corrected the temperature coefficient of this iron loss model through experimental measurement, and verified the correctness of the modified model by testing the silicon steel and motor [17,18]. Some scholars have summarized relevant laws by studying the magnetic properties and mechanical properties of NGO silicon steel within a certain temperature range [19,20,21]. In addition, there are some scholars who also studied the electromagnetic characteristics of oriented silicon steel at different ambient temperatures [22].

In conclusion, more domestic and foreign scholars are paying attention to the characteristics of silicon steel under MPF. However, in the actual motor design process, some designers still rely on the fixed model and basic material data provided by the commercial finite element simulation software and empirical coefficients to design the motor. They still lack the understanding of the nature and change mechanism of the material under the influence of MPF, as well as the final impact on the motor performance.

The first part of this paper summarizes the current research status of the magnetic properties of materials under MPF. In the second part, a high-power density drive motor is designed, and three main factors that affect the characteristics of soft magnetic materials under extreme operating conditions are proposed, which are temperature, stress and frequency. In the third part, a magnetic characteristic test method based on “temperature stress electromagnetic” three-field coupling is proposed, which can proximately simulate the real situation of electrical steel when it runs inside the motor. In the fourth part, 30ADG1500 is used as the experimental object to find out the dynamic characteristics of NGO silicon steel under MPF, and the motor’s efficiency is simulated and compared when the influence factors are not considered and when the influence factors are considered. The rationality of the method is verified through the test result of the prototype and the calculation accuracy is improved by this method. Finally, the conclusion is summarized.

## 2. Motor Design and Multi-Physics Coupling Field Analysis

In order to explore the influence of magnetic characteristics of silicon steel on motor performance, we have designed a new type of high-power density motor based on the actual requirements of engineering, so that electric vehicles can maintain good performance under starting, climbing and braking conditions. Based on the consideration of the motor’s performance using flux weakening speed regulation method and the prototype manufacturing capability, we chose the interior permanent magnet rotor, which adopts the design of 48 slots and 8 poles, and the peak speed of the motor can reach 9000 rpm. Detailed parameters of the motor are shown in Table 1.

In order to further explore what kind of MPF environment the core will be in when the motor is running at high speed, the finite element simulation of the motor was carried out by Motor-CAD and Ansys software. Based on the actual working conditions of electric vehicles, we set the ambient temperature as 60 °C, and predicted the temperature rise and magnetic density of the driving motor at the peak speed, as shown in Figure 1.

When the motor speed reaches 9000 rpm, the magnetic bridge of the rotor will bear about 200 MPa of centrifugal force, which belongs to tensile stress for silicon steel and will affect its loss characteristics. In addition, the fit between the stator and the casing adopts the interference method. According to the interference amount, the calculation shows that the stator will bear about 60 MPa of compressive stress, which will also affect the magnetic characteristics of the silicon steel. Therefore, in the simulation analysis and calculation of the model, we divided the model into four areas according to the actual situation, and applied the characteristic data of silicon steel in different states. The purpose is to improve the accuracy of the calculation.

The color of the magnetic density indicates the degree of saturation of the silicon steel material used in the motor. The red area will heat slightly more than the other areas. As can be seen from Figure 1, the magnetic density of the stator teeth is from 1.8 T to 2 T, the magnetic density of the yoke is mostly between 1 T and 1.2 T, and the average temperature of the iron core is about 120 °C at the peak rotating speed. The magnetic density of most parts of the rotor are from 1 T to 1.4 T, and its maximum temperature is about 100 °C, which is concentrated around the permanent magnet.

In the production process of the motor, the casing and stator usually fixed by interference fit, and the rotor material will be affected by centrifugal force when rotating. Considering those affects, area A of the rotor core is in the physical field of 100 °C and 0 MPa, area B is in the physical field of 100 °C and 200 Mpa of tensile stress (TS), area C of the stator core is in the physical field of about 130 °C and 0 MPa of tensile stress and area D is in the physical field of 125 °C and 60 MPa of compressive stress (PS). Therefore, in order to improve the calculation accuracy, it is necessary to consider the magnetic properties of materials under MPF.

## 3. Experiments under Varying Ambient Conditions

In the harsh aerospace environment such as extremely low temperature, high temperature or vacuum, the parameters of permanent magnet synchronous motor vary greatly, and silicon steel’s characteristic is also prone to nonlinear changes. The coupling relationship among electromagnetic, temperature, fluid, stress and other fields of MPF cannot be ignored. How to accurately test the magnetic properties of silicon steel under MPF and speculate on the motor performance has become a key technical problem.

### 3.1. Test Platform

In this paper, a test system for magnetic properties of silicon steel based on the MPF environment of “temperature, stress and electromagnetic” is established. The test system is composed of three core modules: electromagnetic characteristic test module, mechanical characteristic test module and adjustable test environment module. The magnetic testing equipment adopted a commercial equipment on the market. In addition, in order to test the characteristics of the sample under more applicable conditions, we also customized a tension machine and a device that can adjust the temperature environment, which uniquely coupled the three parts together. This system can not only test the properties of silicon steel under SPF, but also complete the properties of silicon steel under MPF.

As shown in Figure 2, the test platform can simultaneously change temperature, stress, frequency, magnetic field strength and other major variables. We cut the size and shape of the sample according to the relevant national standards, and wrapped the copper wire around the left and right sides of the sample. During the experiment, the sample is fixed in the equipment with adjustable temperature by the tension machine, and the copper wire is drawn to the magnetic characteristics test equipment. In this way, we can test the magnetic properties of the sample, for example, BH and BP data. The design of the test system not only innovatively coupled multiple devices together to provide different environments for silicon steel testing, but also saved space.

When the temperature field is applied, the sample is heated by air heat conduction, and the temperature is fed back by the sensor and adjusted in time. The temperature of the object under test can be varied by contact-free heating or cooling in hot or cold air, so as to avoid interference as much as possible and reduce magnetic leakage of the sample during the test. By applying different voltage, current, frequency and magnetic field intensity to the sample of a commercial magnetic testing equipment of silicon steel, we can obtain the characteristics data of silicon steel samples under different electromagnetic fields. Coupled together, the three devices can be used to load different stress, temperature, electromagnetic and other environments, providing a wider range of support for the magnetic properties of silicon steel testing. Therefore, we can guide the material selection and optimal design of the motor.

### 3.2. Function of Testing Device

Through the thermal simulation analysis of the motor model based on the Motor- CAD software above, it is not difficult to find that the core will be in the environment of high temperature, high frequency and high stress. Therefore, the magnetic properties of silicon steel under multiple physical fields can more accurately reflect the performance of the motor at high speed. Based on this, the following four variables are set in this paper, and 30ADG1500 NGO silicon steel is used as the research object to analyze its magnetic properties. The test range of the test device in this paper is shown in Table 2 below:

## 4. Analysis and Prototype

### 4.1. Analysis of Magnetic Properties of Materials

#### 4.1.1. Magnetic Properties of Silicon Steel Considering Temperature, Stress and Frequency

It was found that the magnetic properties of silicon steel are affected by both stress and temperature. The stress has an effect on the microstructure of silicon steel, which leads to the difference of hysteresis coefficient of different frequencies in the calculation of iron loss. In addition, temperature affects the calculation coefficient of the eddy current loss at different frequencies. Therefore, for the characteristics of silicon steel under multiple physical fields, we need to consider the influence of stress, temperature and frequency on magnetic characteristics of silicon steel for calculation and analysis. Refer to the following formula for calculating iron loss:(1)$$P_{Fe}=P_{h}+P_{c}=k_{h}\left(\sigma \right)fB_{m}+k_{c}\left(T\right)f^{2}B_{m}^{2}$$
$k_{h}$ and $k_{c}$ are, respectively, the hysteresis loss coefficient and eddy current loss coefficient after data fitting.

In order to clearly express the influence of temperature, stress and frequency on the magnetic properties of silicon steel, this paper selected the experimental results under the magnetic field strength of 1000 A/m to draw curves., as shown in Figure 3. In order to show the magnetic properties of silicon steel with the change in stress and temperature, it is drawn as a function of temperature and two-way stresses.

It can be seen that the magnetic density of NGO silicon steel will decrease with the increase in temperature. Compressive stress will also reduce the magnetic density of materials. However, the magnetic density first increases and then decreases with the increase in tensile stress, and reaches a maximum value near 30 MPa. In addition, the magnetic density is more affected by tensile stress at low temperature; with the increase in temperature, the influence of tensile stress on magnetic density decreases gradually.

When the motor is running at a rated speed, the frequency is about 400 Hz. So, the loss of 30ADG1500 under 1 T and 400 Hz was tested, and the changing trend is shown in Figure 4.

It can be seen that the loss of NGO silicon steel will decrease with the increase in temperature; with the increase in applied stress, it first decreases and then increases, reaching a minimum value around 30 MPa. The trend of loss change is opposite to that of the magnetic density change. At high temperature, the loss is greatly affected by tensile stress, and the influence of tensile stress decreases with the decrease in temperature. However, the compressive stress will make the material loss deteriorate to a large extent, and the deterioration extent will gradually increase with the increase in compressive stress.

As the motor’s speed increases, the frequency of the motor increases gradually. When the motor operates at peak value, the operating frequency rises to 800 Hz. In Figure 5, the changing trend of silicon steel’s loss with temperature and stress at 1.2 T and 800 Hz was plotted. The changing trend of loss with stress and temperature is generally consistent with Figure 4, but the turning point of tensile stress of loss trend is reduced from 30 MPa to 25 MPa.

From the experimental data above, at different temperatures, their magnetic properties will show different levels of sensitivity to tensile stress. For example, at low temperature, the iron core’s magnetic density is more affected by tensile stress, while the loss is less affected; with the increase in temperature, the influence of tensile stress on magnetic density decreases and the influence of tensile stress on loss increases.

#### 4.1.2. Comparison of Magnetic Properties of Materials

Figure 6 shows the comparison of magnetic properties of the NGO silicon steel with and without considering MPF.

From the above figure, it can be seen that the magnetic properties of silicon steel will change under the action of temperature, stress, frequency and other external fields. In conclusion, in the future design process of high-performance motors, the calculation accuracy of the model will be greatly improved by considering the magnetic characteristics of silicon steel under MPF.

### 4.2. Analysis of Simulation Results

In this subsection, we replace the A, B, C and D regions of the motor model with the corresponding magnetic characteristics data under high temperature and stress, and compare the motor simulation characteristics of silicon steel under the condition of considering MPF with that of not considering MPF. Based on the data from the test, motor’s loss diagram and efficiency comparison diagram are drawn.

Figure 7 shows the analysis of the magnetic characteristic data for 1 T and 400 Hz of the core material tested in the previous article under MPF. At 4775 rpm, motor’s iron loss considering single electromagnetic field is 233.46 W, and iron loss considering the coupling of MPF including electromagnetic, stress and temperature is 268.29 W. The iron loss of the motor has increased by 34.83 W, which is about 14.9%. At the speed of 9000 rpm, motor’s iron loss considering single electromagnetic field is 466.92 W, and iron loss considering MPF is 536.58 W, the iron loss of the motor has increased by 69.66 W, which is about 14.92%.

With the increase in rotating speed, the proportion of copper loss decreases and the proportion of iron loss increases, and the influence of the characteristics of iron core material on the motor performance increases gradually. Therefore, for the design of high-speed motors, considering the characteristics of iron core materials under MPF can not only more truly reflect the motor performance, but also greatly improve the calculation accuracy.

The loss of the motor during operation is mainly composed of iron loss, copper loss and mechanical loss. For the high-speed motor, efficiency as a very important index, iron loss plays a more important role. For example, for electric vehicles, the high efficiency of the motor can bring about an increase in the driving range. Therefore, we need to analyze the actual loss of the motor at high speed with the help of the magnetic characteristics data of silicon steel under MPF. In order to verify the reliability of the previous test, we compared and analyzed the efficiency of the motor under SPF and MPF, and drew the efficiency graph pair in the form of contour lines, as shown in Figure 8. The numerical statistics of the efficiency are shown in Table 3.

When the motor runs at the peak speed, the stator and rotor core are in a high-temperature state, and the influence of tensile stress on the loss of silicon steel sheet will increase. It can be seen from the table and contour diagram that for the area where the motor efficiency is greater than 95%, the calculated value of motor efficiency considering the characteristic data of silicon steel in SPF is 26.21%, and the calculated value of motor efficiency considering the characteristic data of silicon steel in MPF is 16.67%. The ratio of the area of the high-efficiency area to the total area decreased by 9.54%. In addition, the maximum efficiency of the motor decreased by 0.31%.

Therefore, in the process of motor design, if only the magnetic characteristics of iron core materials under a SPF are considered, the efficiency of motor simulation will be a bit high, which may possibly lead to the problem of insufficient performance after prototype manufacturing, and have a certain impact on the accuracy of motor design. Therefore, in the design of a high-performance motor, MPF has great practical value for improving the accuracy of motor simulation calculation.

### 4.3. Prototype Testing

To verify the work of this paper, a 20/40 kW motor prototype was installed on a 0–15,000 rpm test bench for testing, as shown in Figure 9.

#### 4.3.1. Temperature Rise Comparison

The load testing was carried out on the prototype with the flow of cooling water was 15 L/min, and the inlet temperature was 65 °C. The test results were compared with the simulation results as Figure 10.

As shown in Figure 10, Under the working condition of 30 s peak operation, the simulated temperature rise is about 55 °C and the measured temperature rise is about 62 °C. Different silicon steels usually have different coatings and will also have different levels of sensitivity to temperature rise. After the research, it was found that the coating of the silicon steel we used would make the temperature rise of the core slightly higher than that of other silicon steels. In addition, other non-standard processes in the production process will also have errors in simulation and testing. Therefore, we should strive to improve the accuracy of processing.

#### 4.3.2. Efficiency Comparison

The efficiency simulation data considering MPF were compared with the measured efficiency data as Figure 11.

It can be seen from the Figure 11 that the simulation data under the consideration of MPF is more realistic. Although the area of the high-efficiency area of the motor has been reduced, the total error is within 3% and the maximum efficiency point error is within 0.5%. In addition, in the actual production process of the motor, due to other problems such as process and coating, the efficiency of the motor will also decline slightly.

## 5. Discussion

At present, most motor designs are calculated based on material test data at 50 Hz, 0 MPa and room temperature. However, for future high-performance and special motor requirements, more careful design and precise research under extreme working conditions are still needed. For this reason, the research significance of this paper is to improve the accuracy of model calculations under complex working conditions, and to explore the magnetic characteristics and mechanical properties of silicon steel under MPF from the perspective of material properties testing.

In the experiments carried out in this paper, the main factors such as high temperature, stress and frequency were considered. In the next phase of the test work, we will optimize the function of the test equipment and complete the material characteristics test at extremely low temperature.

## 6. Conclusions

Based on the analysis of a high-power density motor of an electric vehicle, this paper used 1 T and 400 Hz and 1.2 T and 800 Hz working conditions as examples to explore the characteristics of 30ADG1500 iron core material under the MPF of electromagnetic, stress and temperature fields. A summary of our findings is as follows:

1. The magnetic density decreases with the increase in temperature and compressive stress, and firstly increases and then decreases with the increase in tensile stress. The influence of tensile stress on magnetic density at low temperature is greater than that at high temperature.

2. The iron loss of materials will decrease with the increase in temperature; With the increase in tensile stress, the iron loss firstly decreases and then increases, and the influence increases at high temperature. The compressive stress will only worsen the iron loss situation.

In the process of motor design, with the increase in motor speed, the proportion of iron loss is increasing. The magnetic characteristics of iron core materials play a vital role in motor performance. Based on the experiment, it was found that the iron loss of the motor considering the MPF will increase by around 14.9%, leading to a decrease in the actual efficiency of the motor. The range of efficiency greater than 95% will decrease by 9.54%. Finally, the validity of the conclusion is verified by the prototype test result. Therefore, when designing high-performance motors, it is necessary to consider the characteristics of iron core materials under MPF, such as electromagnetic, stress and temperature fields, to improve the calculation accuracy of the model and increase the reliability of motor design.

## Figures and Tables

**Figure 1 materials-15-08305-f001:**
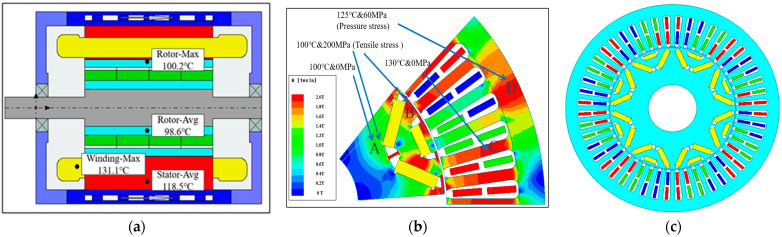
Temperature distribution and magnetic density distribution @ 9000 rpm. (**a**) Temperature rise simulation (flow rate: 15 L/min); (**b**) finite element simulation; (**c**) finite element model.

**Figure 2 materials-15-08305-f002:**
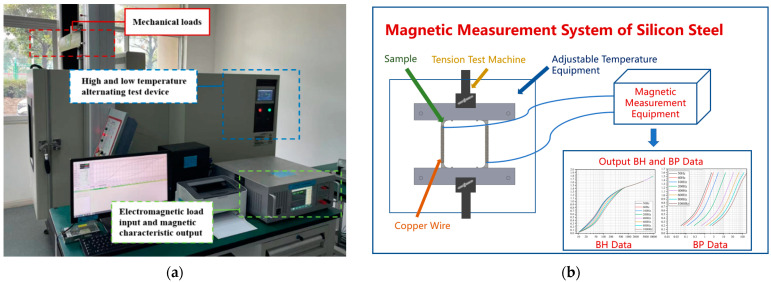
Test platform. (**a**) Actual test system; (**b**) schematic diagram.

**Figure 3 materials-15-08305-f003:**
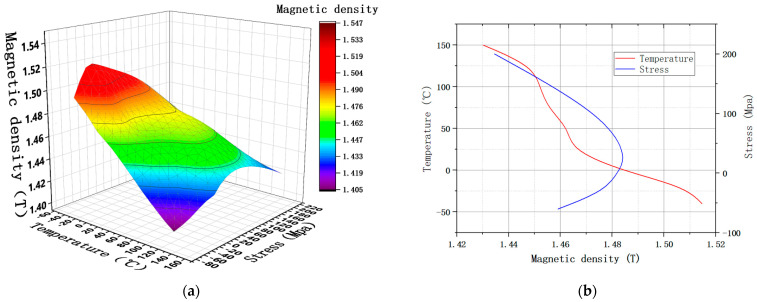
Magnetic density under 1000 A/m. (**a**) The trend of B under MPF; (**b**) B under SPF.

**Figure 4 materials-15-08305-f004:**
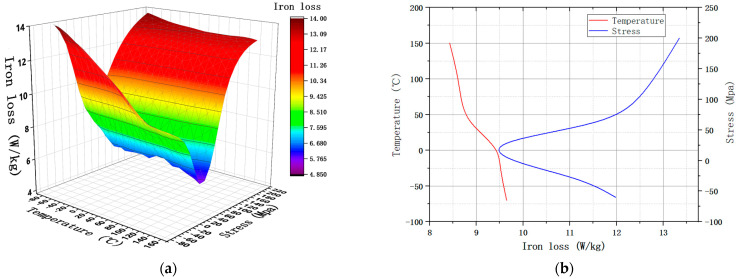
Iron loss under 400 Hz and 1 T. (**a**) The trend of P under MPF; (**b**) P under SPF.

**Figure 5 materials-15-08305-f005:**
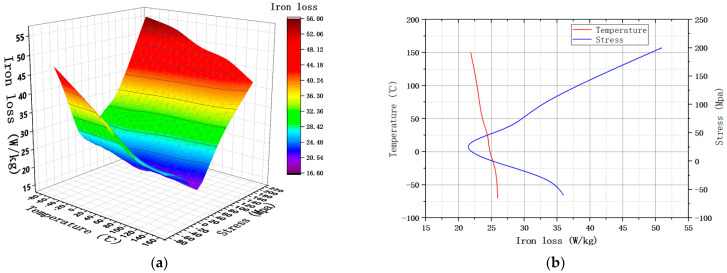
Iron loss under 800 Hz and 1.2 T. (**a**) The trend of P under MPF; (**b**) P under SPF.

**Figure 6 materials-15-08305-f006:**
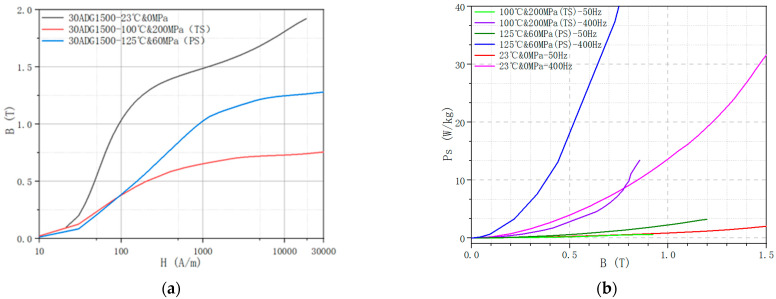
The magnetic characteristic of NGO silicon steel. (**a**) B-H curves; (**b**) B-P curves.

**Figure 7 materials-15-08305-f007:**
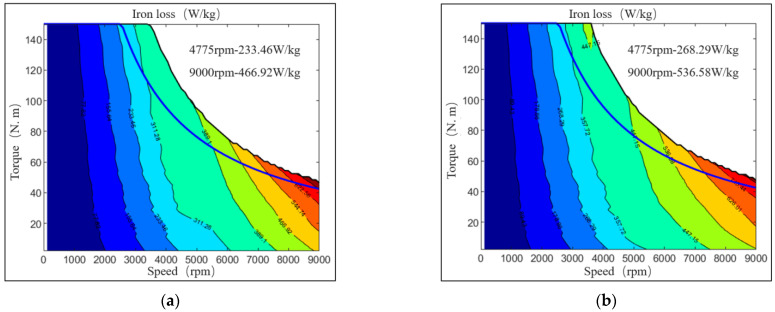
Comparison of motor’s iron loss map. (**a**) Iron loss of SPF; (**b**) iron loss of MPF.

**Figure 8 materials-15-08305-f008:**
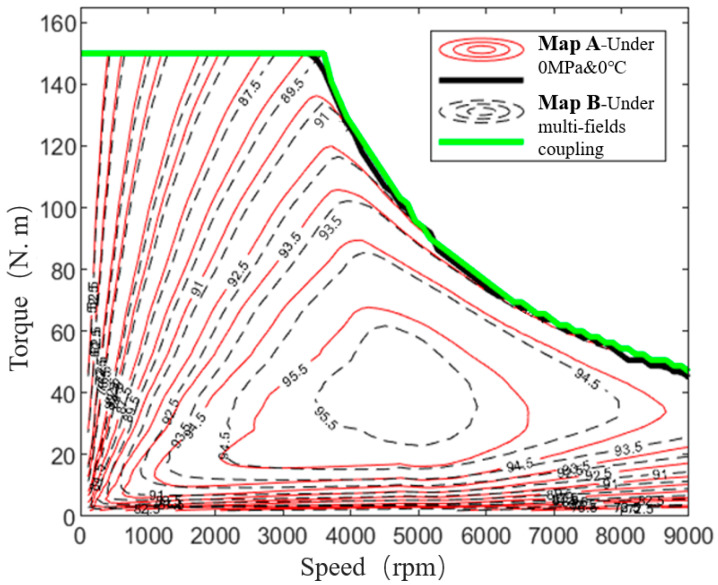
Motor’s efficiency map comparison.

**Figure 9 materials-15-08305-f009:**
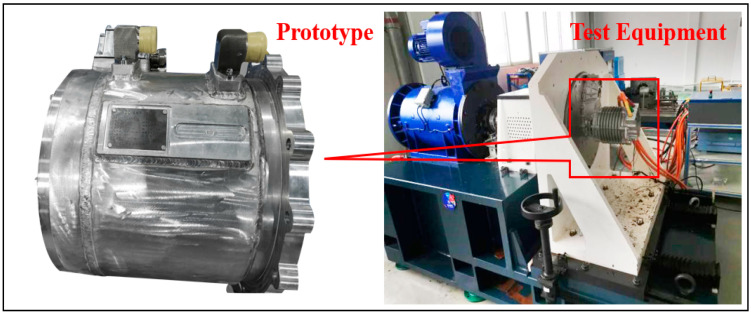
Prototype and test equipment.

**Figure 10 materials-15-08305-f010:**
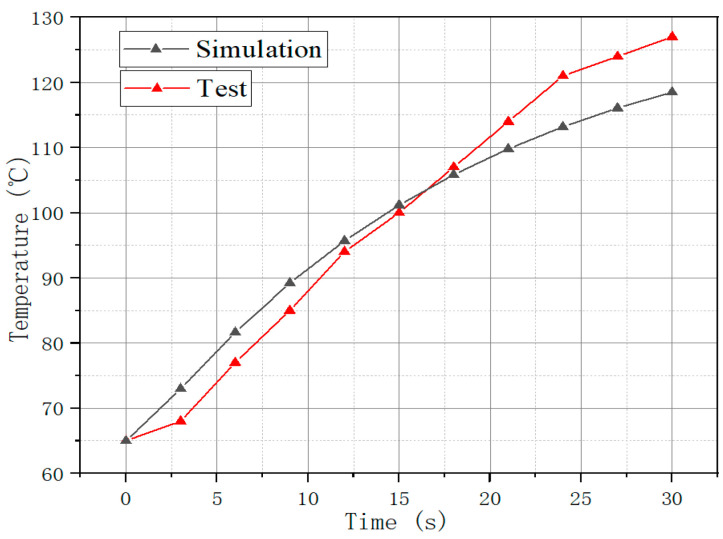
Temperature comparison between simulation and prototype test.

**Figure 11 materials-15-08305-f011:**
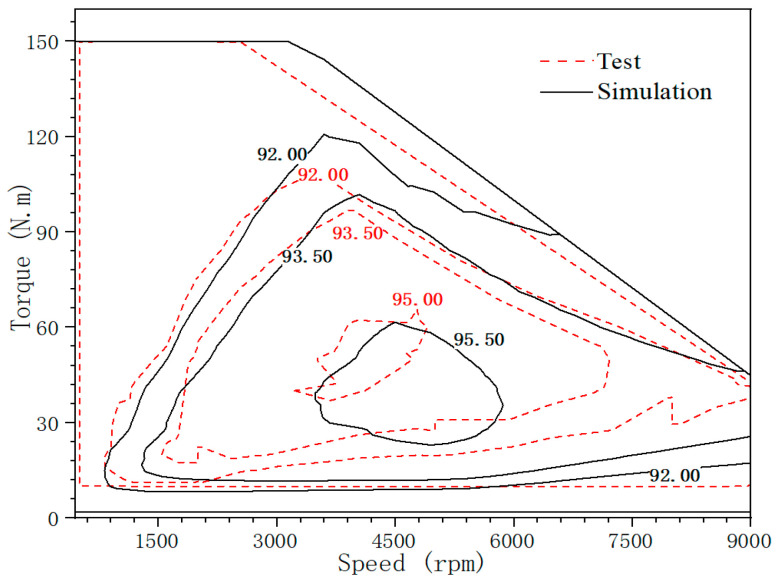
Temperature comparison between simulation and prototype test.

**Table 1 materials-15-08305-t001:** Detailed parameters of the motor.

Parameter	Value
Rated/Peak torque (Nm)	40/150
Rated/Peak power (kW)	20/40
Rated/Peak speed (rpm)	4774/9000
Stack length (mm)	144
Working temperature (°C)	−40~60

**Table 2 materials-15-08305-t002:** Test range of the testing device.

Parameter	Range
Stress	−60–200 MPa
Temperature	−40~150 °C
Frequency	40~1000 Hz
Magnetic field strength	0~10,000 A/m

**Table 3 materials-15-08305-t003:** Comparison of efficiency ratios under two working conditions.

Efficiency	SPF	MPF
Area of efficiency >85%	84.33%	82.55%
Area of efficiency >90%	71.83%	68.38%
Area of efficiency >95%	26.21%	16.67%
Maximum efficiency	96.29%	95.98%

## Data Availability

The data are not publicly available for privacy reasons. The data presented in this study are available from the corresponding author.

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
