# Peer review of "Transient Magnetic Properties of Non-Grain Oriented Silicon Steel under Multi-Physics Field"

_materials, 2022, doi:10.3390/ma15238305_

Round 1

Reviewer 1 Report

In the paper, simulations and experiments of the magnetic and thermal properties of non-oriented silicon steel under varying mechanical stress, temperature and electromagnetic field at variable frequency are presented. The aim of the work is find a parametrized description of these material properties in order to facilitate the design of electric motors for a large working temperature range, as, e.g., required for airborne applications.

I think that the title of the paper could be approved, see my comment 1) below. The abstract summarizes the approach and the main results. In the introduction, the work is put into proper context to previously published research. I think that the sections on “Motor design” and on “Experiments” are much too short and incomplete. With the information given, it is impossible to comprehend what was exactly done and how. Section 4 on “Analysis and prototype” is much better, it nicely illustrates the findings for magnetic field density, iron loss and torque as function of applied stress and temperature. These results are very useful for people designing their own motors and electromagnetic equipment. In the conclusion, the most important results are summarized in brief.

In revision, the following issues need to be addressed:

1) Title: I stumbled over the term “Multi-Physics Coupling Field”. To the best of my knowledge, “Multi-Physics” or “Multiphysics” in general denote simulations that couple multiple physical phenomena, usually from different scientific disciplines, as discussed in “D.E. Keyes et al, Multiphysics simulations: Challenges and opportunities. International Journal of High Performance Computing Applications 27(1) 4–83 (2012), DOI: 10.1177/1094342012468181”. In addition, “Multiphysics” in one word is a trademark of Comsol Multiphysics company, naming their general-purpose simulation software. I have never seen or heard Multiphysics in combination with “Coupling Field”. It should be clarified if you mean “Multiphysics Simulations” in general or “Comsol Multiphysics” in particular. In the former case, I suggest to change the title to “Multiphysics Simulations and Experimental Study of Transient Magnetic Properties of Non-Oriented Silicon Steel”, in the latter case to “Comsol Multiphysics Simulations and Experimental Study of Transient Magnetic Properties of Non-Oriented Silicon Steel”. In fact, I suggest to delete “Temperature-Stress-Electromagnetic” from the title, in order to keep the title somewhat shorter.

2) Abstract, line 14: in continuation of my concerns about the title, I suggest to replace “multi-Physics coupling field (MPCF)” by “Multiphysics simulations (MPS)”. Correspondingly, the term “single physical field (SPF)” could be replaced by “single physical quantity (SPQ)”.

3) Introduction, lines 43-45: you wrote: “The magnetic properties and mechanical properties of silicon steel under MPCF have gradually become a research hotspot.” I suggest to give some evidence, for instance references to some recent works.

4) Section 2 “Motor Design …”: you wrote in lines 97-98: “… a new high-power density drive motor for electric vehicles is designed in this paper.” However, I completely missed information on how the design was performed. Just some motor parameters and some values and distributions of resultant temperatures and magnetic fields are given. It has to be mentioned which software was used for the simulation. Was it Comsol multiphysics, or some other software? What were the design considerations and constraints? The required temperature range is mentioned to be between –40°C and +60°C, but it is not mentioned which ambient temperature was assumed for the simulated temperature distribution depicted in Figure 1 (a). I am sure that the temperatures of the different motor components will strongly depend on the ambient temperature. I understand that the design process is not the main focus of the paper. Section 2 could just serve as a justification of the necessity to perform multiphysics simulations with stress- and temperature-dependent magnetic properties of the steel components of the motor. In that case, it would not be necessary to present details of motor design. It would suffice to present the simulation results without a lengthy discussion of the design process. However, if you write in line 98: “.. is designed in this paper ..” and in line 105: “.. the finite element simulation is carried out in this paper ..”, then the design needs to be described in detail.

5) Section 3 “Experiments” (line 123): I suggest to rename the title of the section to “Experiments under varying ambient conditions” or similar.

6) Section 3.1 “Test platform” (lines 132-156): I think the information that is given on the test platform is too scarce. It should be mentioned if it is (in total or in part) a commercial test platform or if it is a custom-built individual apparatus. In the former case, the manufacturer(s) and the equipment name(s) needs to be given. In the latter case, you should elaborate in much more detail on the design of the test platform.

7) Line 149: you wrote “Both high and low temperature field is conducted without contact”. I think you mean “The temperature of the object under test can be varied by contact-free heating or cooling in hot or cold air.” or similar.

8) Line 151: you wrote “When loading electric fields, …”. I think it is not correct. I think you apply magnetic fields at variable frequency, don’t you? Please explain the “magnetic property tester”. Is it a commercial device, or is it home-built? How does it work? I think this is a key component of your test platform, because the magnetic properties of the steel are extracted from that device.

9) Figure 8, Table 3 and text lines 235-253: I must admit that I do not understand this section at all. What efficiency values and efficiency ratios are actually discussed here? Is it (as usual and as implied by Figure 8) output mechanical power divided by input electrical power? Why do the values listed in Table 3 suddenly drop to 26% or even 16%. What do you mean in lines 244-245 with: “For the range which efficiency greater than 95%, the test data considering the material characteristics under a SPF is 26.21%”. Which “test data”? Sorry, I do not understand at all. Please re-write this section.

10) Section “References” (lines 332-376): please format all references correctly as required by MDPI Materials (https://www.mdpi.com/authors/references). In particular, please delete all phrases “Author 1” and “Author 2”. Please list all authors of each reference (unless more than 10 authors, then use “et al.”). I think you used IEEE citation style. Please change it to MDPI citation style. Please also give DOI of all publications (if available).

Author Response

Dear Editors and reviewers:

Thank you for your precious comments and advice. Those comments are all valuable and very helpful for revising and improving our paper, which have important guiding significance to our researches. We have read the comment carefully and have made correction which we hope meet with approval. The corrections we made and the responses to the reviewer’s comments are as flowing:

Response to Q1:

We have taken your suggestion and changed the title to: Transient Magnetic Properties of Non-grain Orientated Silicon Steel under Multi-physics Field.

Response to Q2 and Q3:

Under your suggestion, we have revised the abstract and introduction.

Response to Q4:

Thanks for your valuable advice, we add a finite element model of the electric motor to the manuscript. In our previous study, we made a complete design and check, including thermal simulation, electromagnetic simulation and rotor strength simulation. In the new manuscript, we added more detailed design parameters.

Response to Q5, Q6, Q7 and Q8:

First of all, our team would like to express our thanks for your careful examination. Your suggestions have provided great help to the revision of this manuscript. The test system consists of a commercial magnetic properties test equipment and two non-standard customized equipment. In the new manuscript, we published as much information about the test system as possible.

Response to Q9:

Efficiency is an important parameter for evaluating motor performance, which is directly related to silicon steel materials. In order to reflect the performance of the motor more directly, we draw the simulation results of the motor efficiency into a contour map. We apologize for the lack of clarity in the original manuscript. We adopted to your suggestions and made the relevant changes.

Response to Q10:

We have modified the format of the references according to the information you provided.

Thank you for your careful review. We really appreciate your efforts in reviewing our manuscript. We wish good health to you, your family, and community. Your careful review has helped to make our study clearer and more comprehensive.

Reviewer 2 Report

Major:

Please insert a schematic diagram of the test platform of figure 2 to better understand the experimental procedure.

Minor:

Page 8, row 222, put a comma after 7 not a dot

Page 10, the first sentence of conclusions should be canceled, it is probably a part of the template.

Author Response

Dear Editors and reviewers:

Thanks very much for taking your time to review this manuscript. I really appreciate all your comments and suggestions! Please find my itemized responses in below:

Response to Q1:

We have updated the schematic diagram of the experimental equipment in the manuscript.

Response to Q2 and Q3:

We apologize for the errors in the original manuscript and have revised it.

Thank you again for your careful review. We really appreciate your efforts in reviewing our manuscript. We wish good health to you, your family, and community. Your careful review has helped to make our study clearer and more comprehensive.

Reviewer 3 Report

The combination of calculations using the Maxwell electrodynamics equations with the N-S flow equations, and their impact on the results of structural simulations (multi-Physics coupling field (MPCF)), is the highest level of engineering issues. It proves the excellent skills of the Authors and certainly places the works in hi-tech trends.

"The first part of this paper summarizes the current research status of the material's magnetic properties under MPCF. In the second part, a high-power density drive motor is designed, and three main factors that affect the characteristics of soft magnetic materials under extreme operating conditions are proposed: temperature, stress, and frequency. Finally, a magnetic characteristic test method based on "temperature stress electromagnetic" three-field coupling is presented in the third part. "

One of the drawbacks of the work is the lack of any equations describing the fields mentioned above and closures (constitutive equations) connecting them. The authors designed the electric motor using the material characteristics obtained in the coupled area of interaction. 

Several linguistic inaccuracies were found in the work that requires correction.

12 row "traditional motors is" - (better is) traditional engines is

13 "al- ready high enough, however, for the future demand of high-performance"  -> (better is) al- ready high enough; however, for the future demand of high-performance

35  "Similarly, the heat dissipation problem and demagnetization problem caused by temperature rise during the operation of permanent magnet  synchronous motor indicate that it is necessary to consider the influence of frequency,  stress and temperature on the core material." 

Hence my attention to providing the equations describing the impact of the fields mentioned above on the degradation of the surface or the core of the material.

106. "Fig.1 Temperature distribution and magnetic density distribution under peak working condi-tion. (a) Temperature distribution; (b) Magnetic density distribution."

The figure shows a diagram of the motor cross-section, a) and the magnetic flux field, B. The temperature and stress fields were limited to the determination of point values. If a project is in the title, then there should be separate fields, or a subitem should be corrected.

107 No diagram shows boundary conditions and loads for thermal analysis. In addition, there is no data on the validation of the model. Showing the spatial fields in cross-sections or parts would significantly improve the clarity of the work. 

108. The magnetic density in most part of the rotor is 1T - 1.4T, and its maximum temperature is about 100 C -->(better is) The magnetic density in most part of the rotor is 1T to  1.4T, and its maximum temperature is about 100 C.

128 The coupling relationship between the electromagnetic field, temperature field, fluid field,  stress field, and other physical fields is more complex, and the coupling relationship between MPCF cannot be ignored. --> (better is)   The coupling relationship between electromagnetic, temperature, fluid, stress and other fields of MPCF cannot be ignored.  

146  Place the sample in an insulated temperature box and fix it with a clamp connected to the mechanical load module. (to correct the time for past)

178 The authors describe changes in the magnetic flux density as a function of temperature and two-way stresses. Can they explain the reasons why the graph takes this shape?

182 Fig.2 How are losses defined? Unit W/kg describes the energy transmission loss or the weight loss in the material kg/m3?

217 "Using the data tested, motor’s loss diagram and efficiency comparison diagram are drawn." -->use past time

266.  Due to the coating of silicon steel, the difference of temperature rise will be brought. Therefore, the actual effect of temperature on the material will increase slightly. --> not cleary sentence. 

292 "This section is not mandatory but can be added to the manuscript if the discussion is  unusually long or complex." ??? This sentence is taken from the author's guide. 

301 "The loss of materials will decrease with the increase of temperature;" The loss of resistance/conductivity or material mass?  What is the degradation mechanism if it is a loss of mass material? 

Thanks.

Author Response

Dear Editors and reviewers:

Thanks very much for taking your time to review this manuscript. I really appreciate all your comments and suggestions! Please find my itemized responses in below:

Response to Q1:

Through the test using our customized test platform, our team found that the iron loss and magnetic density test results of silicon steel materials under multi-physical fields are different from those under single physical field. Based on influencing factors such as temperature and stress, we have revised the coefficients for calculating iron loss.

Response to Q2: line 12 to line 301

Thank you for your valuable suggestions. We have finished revising the manuscript. The new manuscript looks more logical than before. In response to your doubts, I would like to explain a few points to you.

(1) The magnetic characteristics measurement equipment is used to reflect these characteristics in a numerical way. We usually use W/kg as a unit to express the amount of iron loss.

(2) In order to show the change trend of silicon steel’s magnetic properties under different stress and temperature, a function which considered temperature and two-way stresses is added.

Thanks again for your careful review. We really appreciate your efforts in reviewing our manuscript. We wish good health to you, your family, and community. Your careful review has helped to make our study clearer and more comprehensive.

Round 2

Reviewer 1 Report

The paper has been significantly improved in revision. All reviewers’ suggestions have been addressed in detail. I like the new title of the paper much better than before. An additional reference on the recent importance of research on the magnetic and mechanical properties of silicon steel has been added. The section on motor design has been largely rewritten, adding more information on how the design was performed. The procedure of design and verification including thermal simulation, electromagnetic simulation and rotor strength simulation is presented. Information on the temperature of the simulations, +60°C, has been added. The title of section 3 has been changed as suggested. In-depth information on the test platform including a detailed description on which component is commercial and which is custom-built has been included in the text. A precise description of the temperature variation procedure has been added. The functionality of the testing apparatus with its ability to apply variable stress, temperature, electromagnetic fields and other environmental conditions is emphasized. Thus, the testing platform provides a wide range of ambient conditions for examining the magnetic properties of silicon steel. The description of evaluating motor efficiency has been rewritten. Now it is much clearer to me. The simulation results of the motor efficiency are discussed in more depth and are nicely visualized graphically. The formatting of references has been corrected according to the journal’s requirements. In summary, I think the paper has been improved sufficiently in revision to warrant publication.